# Non-Coding RNAs as Potential Diagnostic/Prognostic Markers for Hepatocellular Carcinoma

**DOI:** 10.3390/ijms252212235

**Published:** 2024-11-14

**Authors:** Federica Tonon, Chiara Grassi, Domenico Tierno, Alice Biasin, Mario Grassi, Gabriele Grassi, Barbara Dapas

**Affiliations:** 1Clinical Department of Medical, Surgical and Health Sciences, Cattinara University Hospital, University of Trieste, Strada di Fiume 447, 34149 Trieste, Italy; effetonon@gmail.com (F.T.); domenico.tierno@units.it (D.T.); 2Degree Course in Medicine, University of Trieste, 34127 Trieste, Italy; chiara.grassi2@studenti.units.it; 3Department of Engineering and Architecture, University of Trieste, Via Valerio 6, 34127 Trieste, Italy; al-ice.biasin@phd.units.it (A.B.); mario.grassi@dia.units.it (M.G.); 4Department of Chemical and Pharmaceutical Sciences, University of Trieste, Via L. Giorgieri 1, 34127 Trieste, Italy; bdapas@units.it

**Keywords:** hepatocellular carcinoma, miRNA, lncRNA, circRNA, prognosis, diagnosis

## Abstract

The increasing incidence of hepatocellular carcinoma (HCC), together with the poor effectiveness of the available treatments, make early diagnosis and effective screening of utmost relevance. Liquid biopsy represents a potential novel approach to early HCC detection and monitoring. The identification of blood markers has many desirable features, including the absence of any significant risk for the patients, the possibility of being used as a screening tool, and the ability to perform multiple tests, thus allowing for the real-time monitoring of HCC evolution. Unfortunately, the available blood markers for HCC have several limitations, mostly related to specificity and sensitivity. In this context, employing non-coding RNAs (ncRNAs) may represent an interesting and novel diagnostic approach. ncRNAs, which include, among others, micro interfering RNAs (miRNAs), long non-coding RNAs (lncRNAs), and circular RNAs (circRNAs), regulate human gene expression via interactions with their target mRNA. Notably, their expression can be altered in HCC, thus reflecting disease status. In this review, we discuss some notable works that describe the use of miRNAs, lncRNAs, and circRNAs as HCC biomarkers. Despite some open aspects related to ncRNA use, the presented works strongly support the potential effectiveness of these molecules as diagnostic/prognostic markers for HCC.

## 1. Introduction

The increasing incidence [1] of liver cancer, together with the fact that it currently represents the third leading cause of cancer deaths worldwide, just after lung and colorectal cancers [2], makes liver cancer a global health challenge [3]. Primary liver cancer is a malignant tumor characterized by different histological types. Hepatocellular carcinoma (HCC), the most common type of primary liver cancer (75–85% of primary liver cancer cases [4]), originates from hepatocytes [5]. The risk of HCC development has marked regional differences [4]: globally, the age-standardized incidence/mortality is highest in Eastern Asia, South-Eastern Asia, and Northern Africa [2]. HCC’s prognosis is very poor, with a 5-year overall survival rate of less than 12% [6]. More than 90% of HCCs occur in the context of chronic liver disease, especially liver fibrosis [7]. In this regard, it is now known that liver fibrosis produces a “premalignant” environment [7] favoring HCC development [8]. Most HCC diagnoses occur in 60/70-year-old-subjects, with men mostly being affected [9]. Alcohol consumption, hepatitis B/C infection, non-alcoholic fatty liver disease, non-alcoholic steatohepatitis, diabetes, cigarette smoking, and exposure to aflatoxins are the most relevant risk factors [5].

The HCC molecular signature varies depending on the etiology. Different molecular pathways, including those regulating the cell cycle, DNA methylation, chromosomal stability, apoptosis, the immune system, and non-coding RNA expression, can be affected [10]. The molecular heterogeneity of HCC makes the identification of effective drugs very challenging.

### 1.1. Available Therapeutic Approaches for HCC

The stage of the disease guides the choice of the most appropriate treatment. For localized forms of HCC (single nodule), surgical resection is indicated for patients with HCC occurring in non-cirrhotic liver or in the case of a limited cirrhotic liver with conserved liver functions [11]. Liver transplantation is indicated for patients bearing one nodule <5 cm in size or up to three nodules each <3 cm in size (Milan criteria [12]). However, the limited availability of healthy donor livers makes this option rarely applicable. To treat intermediate disease, the employment of trans-arterial chemoembolization (TACE) and trans-arterial radio-embolization (TARE) is considered when two to seven nodules, the largest of which is <7 cm in size [13], are present. TACE and TARE can also represent a bridging therapy in patients waiting for liver transplantation [14]. In the advanced stages of the disease, it is possible to consider using targeted therapies and immune checkpoint inhibitors [13]. Systemic therapies can also be used in patients with disease recurrence after a curative intent. First-line systemic therapies based on oral multi-kinase inhibitors such as sorafenib are indicated in patients with contraindications in the use of immunotherapy [13]. If no contraindication to immunotherapy exists, first-line drugs such as atezolizumab and bevacizumab can be used. These are humanized monoclonal antibodies directed against programmed cell death protein 1 ligand (PD-L1) or vascular endothelial growth factor (VEGF-A), respectively. In particular, atezolizumab blocks PD-L1, favoring the anti-HCC effects of T-lymphocytes, while bevacizumab downregulates HCC neo-angiogenesis. Finally, for the terminal stage of HCC, no anticancer treatment can be proposed; only palliative approaches. Unfortunately, about 20/30% of patients worldwide are diagnosed at this end stage of the disease.

### 1.2. HCC Diagnosis

#### 1.2.1. Instrumental Diagnosis

The early detection of HCC is crucial, as the therapeutic options available are more effective for the first stages of the disease. Ultrasound examination (UE) [13] represents an elective diagnostic tool for HCC screening. The sensitivity range of UE is 51–87%, while its specificity is 80–100% [15]. UE can determine the location, size, morphology, and vascular invasion of HCC. Determining the circulating level of alpha-fetoprotein (AFP) can be associated with UE. AFP is a serum glycoprotein produced by the fetal yolk sac and fetal liver during gestation. While its level can increase in the course of HCC, other pathological conditions, such as chronic hepatitis, cirrhosis, pregnancy, and some germ-line tumors, can induce its increase. Furthermore, approximately 40% of small HCCs do not secrete AFP [16]. For these reasons, it is not considered to be particularly useful in terms of cost-effectiveness. However, recent data indicate that combining UE and AFP could increase HCC detection to 63% compared to 45% supported by UE alone [17].

When the liver is difficult to analyze using UE, contrast-computed tomography (CT) is indicated. The sensitivity and specificity of CT are 65% and 96%, respectively [18]. However, for lesions smaller than 2 cm, the sensitivity decreases to 40%.

Finally, magnetic resonance imaging (MRI) [19] can be considered for HCC diagnosis, even if questions arise about the cost of screening and equipment availability.

#### 1.2.2. Biochemical Diagnosis

In the case of lesions smaller than 1–2 cm, which cannot be accurately characterized by means of CT/MRI, a liver biopsy is indicated for HCC diagnosis [11]. In addition, tissue biopsy enables the histological classification of HCC, thus helping to define disease prognosis. Biopsy limitations are represented by sampling bias and tumor heterogeneity, which may result in a tissue sample not fully representative of the HCC histology [20]. Moreover, liver biopsy may be associated with side effects such as pain and bleeding [21]. Notably, so far, HCC has no tissue biomarker-driven therapy, in contrast to other tumors, such as breast and lung cancers [22].

Recently, liquid biopsy has arisen as a potential novel approach to HCC diagnoses [20]. The possibility to detect circulating HCC markers has several advantages, such as the possibility of performing multiple tests, thus allowing for the real-time monitoring of HCC evolution. Moreover, liquid biopsy has no substantial relevant side effects in comparison to liver biopsy. So far, however, the employment of liquid biopsy is not an integral part of the standard clinical practice. This is mainly due to concerns dealing with sensitivity, specificity, and cost-effectiveness. While we believe that sensitivity/specificity issues can be overcome using modern omics techniques, the problem relies on the identification of a robust and specific circulating HCC marker(s). In principle, liquid biopsy has the potential to help in the identification of minimal residual disease, treatment selection, and the monitoring of therapeutic responses. Unfortunately, so far, the only serum makers for HCC detection have been AFP, as discussed above. Novel HCC markers being tested include the fucosylated fraction of AFP (AFP-L3) and des-gamma carboxyprothrombin (DCP) [23].

The novel circulating biomarkers being investigated for use in HCC detection are represented by circulating tumor cells (CTCs), cell-free DNA (CFD), and non-coding RNAs (ncRNAs). The detection of CTCs is one of the first approaches in a liquid biopsy. The concept is that CTCs can migrate from the tumor of origin/metastasis to the vasculature; therefore, their circulating number is somewhat proportional to the tumor burden. While they may not be indicated in detecting early HCC (a reduced number of CTCs), they may be suitable for monitoring HCC recurrence and the treatment response [24]. CFD is represented by a large pool of double-stranded DNA fragments associated with nucleosomes circulating in the blood. Typically, CFD has been studied to determine the quantity, integrity, and copy number alterations [25]; more recently, it has been studied with regard to the methylation pattern and mutational signature. In this regard, different epigenetic changes in the CFD of patients with HCC have been defined [20], possibly contributing to improved diagnostic specificity. ncRNAs are the topic of this review and are presented extensively in the following sections.

#### 1.2.3. Considerations Regarding HCC Diagnosis/Monitoring

As discussed above, many different approaches are available for HCC diagnosis/monitoring; however, almost all of them have limitations that may delay HCC identification and/or make the monitoring less effective than we would like. Liquid biopsy represents a minimally invasive technique with relevant diagnostic potential for HCC and other pathological conditions. Unfortunately, for HCC diagnoses, a robust diagnostic/prognostic marker has not yet been identified. In this regard, it is possible that rather than a single marker, multiple biomarkers must be analyzed to determine an optimal result. Moreover, it is possible that a multi-parametric approach, which includes biomarkers and different patient features, will be necessary for a more effective output compared to biomarkers alone. In this context, we believe that ncRNAs may substantially contribute to reaching the goal of a specific and sensitive HCC diagnostic/monitoring approach.

## 2. Non-Coding RNAs

To date, many different non-coding RNAs (ncRNAs) have been recognized [26]. In addition to the well-known ribosomal RNAs (rRNAs) and transfer RNAs (tRNAs), other types have been identified in recent years [27]. These include micro interfering RNAs (miRNAs), long non-coding RNAs (lncRNAs), circular RNAs (circRNAs), heterogeneous nuclear RNAs (hnRNAs), PIWI-interacting RNAs (piRNAs), small nuclear RNAs (snRNAs), and small nucleolar RNAs (snoRNAs). NcRNAs regulate human gene expression via interaction with their target mRNAs. In addition, they also interact with each other, resulting in a very complex system able to control human gene expression. Thus far, the most studied ncRNAs in relation to human pathologies and to HCC in particular are miRNAs, lncRNAs, and circRNAs, all recognized as essential regulators of several biological processes in health and disease.

### 2.1. miRNAs

miRNAs, first recognized as potential regulators of human gene expression in 2001 [28], are short double-stranded non-coding RNAs. To date, more than 2000 human mature miRNAs have been annotated in the miRNA archive miRBase [29].

miRNAs originate in the cell nucleus (Figure 1) from a long precursor named primary miRNA (pri-miRNA) derived from coding/non-coding RNAs [30].

Following processing by the enzyme Drosha [31], the pri-miRNA is converted into so-called pre-miRNA, which is exported to the cytoplasm by Exportin 5 (Exp5). Here, the DICER enzyme [32] generates a double-stranded RNA (mature miRNA) of approximately 22 nucleotides bearing 2 nt 3′ overhangs [33]. The antisense strand of the mature miRNA is then up-taken by the enzymatic complex RISC (RNA-induced silencing complexes) [32], allowing for the recognition of the target mRNA via a complementary base-pairing approach. A perfect match of the first 2–8 nucleotides, from the 5′ end of the antisense strand with the target, is required [34] for efficient translation inhibition. When full complementarity occurs among all nucleotides, the target RNA is degraded. The above mechanism of interaction, occurring at the 3′UTR of the target mRNA, is considered the classical mechanism of action. However, miRNAs, which can interact with 5′UTR or with the coding region of an mRNA molecule, have also been described [35]. Notably, the biological effect of miRNAs is not limited to the downregulation of gene expression as, in some cases, they can promote gene expression via direct/indirect mechanisms [36].

Together, the above considerations indicate the biological complexity of miRNAs, which is further amplified by the fact that a single miRNA can target several different mRNAs, and many miRNAs regulate the same mRNA [37]. This promiscuous modality of action is further amplified by the fact that miRNAs regulate and are regulated by lncRNAs/circRNAs (see below). Moreover, it is now evident that miRNAs can circulate in the blood carried by extracellular vesicles, acting as hormones, as assumed below [38]. The role of miRNAs in the pathogenesis of HCC, their potential values as novel biomarkers/therapeutic targets [39,40,41,42], and their response to pharmacologic therapy are now well known [43,44].

### 2.2. Long Non-Coding RNAs

LncRNAs are single-stranded RNA molecules longer than 200 nucleotides [45]. The first lncRNAs described in eukaryotes date back to the beginning of the 1990s [46]. Nowadays, the NONCODE database reports 173,112 lncRNA transcripts [47].

LncRNA synthesis (Figure 2) is comparable to that of mRNAs: indeed, lncRNAs are transcribed by RNA polymerase II (Pol II) and are often spliced, capped, and poly-adenylated.

Unlike mRNAs, lncRNAs do not contain a translated open reading frame. Moreover, they are generally shorter and are expressed at lower levels compared to mRNAs. Finally, they usually contain fewer but longer exons than mRNAs, are often retained in the nucleus, and tend to have more tissue-specific expression [48]. They can be transcribed in sense and antisense orientations from diverse genomic regions such as introns, exons, intergenic regions, pseudogenes, telomeres, centromeres, gene promoters, and 3′UTR [49].

The linear nucleotide sequence of lncRNAs determines their 3D structure, which is ultimately responsible for their biological effects [50]. Based on the 3D structure, lncRNAs can bind and recruit transcription activators/repressors to the promoters of their target genes, thus modulating transcription. They can bind miRNAs to complementary miRNA sequences (sponge effect), thus inhibiting miRNAs’ biological effects. Moreover, lncRNAs can serve as scaffolds to promote the formation of protein complexes, thus promoting/inhibiting protein functions. Altogether, these features indicate lncRNAs’ complex and multiple biological functions, making their altered expression and/or mutation implicated in numerous human diseases unsurprising [51]. Currently, the involvement of lncRNAs in HCC with regard to the pathogenesis of the disease and the possibility of using them as biomarkers/targets for novel molecular therapeutic approaches is well documented [40,41,42].

### 2.3. circRNAs

circRNAs have a circular structure constituted by a single-stranded RNA covalently closed at the extremities [52]. The circular shape confers to these molecules increased stability against degradation by exonucleases compared to linear RNAs. Initially described in the early 1990s, we now know that the expression of these molecules is rather wide, as over 10,000 circRNAs have been described in many living organisms, including humans [53].

RNA polymerase II (Pol II) [54] drives the transcription of circRNAs from precursor mRNA (pre-mRNA) (Figure 3).

CircRNAs can derive from exons, introns, exon–intron junctions, or intergenic regions of the genome [55]. Interestingly, the same gene can give origin to multiple circRNAs via an alternative splicing mechanism [56], and often, cirRNA expression is unrelated to the expression of the host gene [57]. CircRNAs are generated in the cell nucleus via a mechanism called back-splicing, which most likely requires canonical spliceosomal machinery [58]. Subsequently, they are exported to the cell cytoplasm. CircRNAs’ length varies, ranging from 30–40 nucleotides to longer sequences [59]. While the expression level is usually low [60], abundantly expressed circRNAs have also been described [61]. To date, different biological functions have been defined for circRNAs [26]. In particular, they can (1) bind miRNAs via complementary regions (sponge effect), thus impairing miRNA effects; (2) interact with specific mRNAs, regulating their stability and/or translation; (3) undergo translation to generate small peptides; (4) interact with RNA-binding proteins acting as decoys or sponges for proteins; and (5) interact with gene promoters, thus modulating gene expression. CircRNAs’ involvement in HCC pathogenesis and their possible role as novel biomarkers/therapeutic targets is commonly accepted [40,42].

## 3. Extracellular Vesicles

The observation that ncRNAs can circulate in the blood embedded into extracellular vesicles (EVs), thus acting as hormones, has disrupted research on the biology of EVs. These are a heterogeneous group of lipid bilayer particles synthesized and secreted by innumerable cell types into the extracellular environment [38]. Thus, they can circulate in many biological fluids, including blood [62]. Based on their size, they can be classified as exosomes (30–150 nm in diameter), ectosomes (50–10,000 nm in diameter), apoptotic bodies (1000–5000 nm in diameter), and other vesicles [63]. The precise molecular mechanisms of EV generation remain partially unclear; however, it is accepted that they form either through inward-budding vesicles within the endocytic system or outward-budding vesicles at the plasma membrane (Figure 4).

The first demonstration of the existence of EVs dates back to 1987 [64], but the biological meaning became clearer in 1996 [65], especially in 2007, when it became evident that EVs can carry cytoplasmic molecules such as mRNA and miRNAs [66]. Afterward, it was observed that both stromal and cancer cells can produce EVs modulating tumor progression through molecular exchange [67]. More recently, it was shown that ncRNAs carried by EVs regulate HCC treatment response and disease progression [68]. Nowadays, it is accepted that circulating exosome-derived ncRNAs, like miR-21 [69], lncRNA ASMTL-AS1 [70], lncRNA TUC339 [71], and circ_0051443 [72], are strongly related to HCC disease.

## 4. ncRNAs as Biomarkers for HCC

### 4.1. miRNA

miRNAs were the first class of ncRNA to be studied as potential diagnostic/prognostic markers for HCC. Among the many works published in the field, here, we focus on a selection of representative papers (Table 1).

#### 4.1.1. miR-221 and miR-18a

miR-221 is encoded on the human X chromosome in humans and regulates target genes essential for the growth of several types of cancers [73]. miR-18a is one of the most conserved and multifunctional miRNAs in the polycistronic miR-17-92 cluster. It is frequently overexpressed in malignancies, such as non-small-cell lung cancer, cervical cancer, and gastric cancer. Among the various recognized targets of miR-221 is the Wnt/β-catenin pathway [74], while miR-18a has been shown to regulate the expression of KLF transcription factor 4 (KLF4) and p21, promoting HCC cell proliferation and migration [75]. Both KLF4 and p21 are inhibitors of the cell cycle.

Yun et al. [76] evaluated the dysregulation of miR-221 and miR-18a in 50 formalin-fixed, paraffin-embedded tissues obtained from surgically resected HCC. Both miR-221 and miR-18a were found to be more expressed in HCC tissues compared to adjacent noncancerous tissues via real-time qRT-PCR. In addition, the survival rate of patients with low miR-221 expression was significantly higher than that of patients with high expression (*p* = 0.020). Although miR-18a expression was not correlated with survival or recurrence, the authors found that miR-18a levels were significantly associated with tumor stage and size.

Chen et al. also suggested the potential role of miR-221 as a prognostic biomarker for HCC [77]. Significant upregulation of miR-221 expression was detected in 135 primary HCC tissues compared to non-tumor tissues using real-time qRT-PCR. Notably, in patients with multiple tumor nodules and microvascular invasion, miR-221 was upregulated compared to patients with a single tumor nodule and no microvascular invasion. In addition, miR-221 levels showed a close association with tumor stage. Kaplan–Meier survival analysis revealed that HCC patients with low miR-221 expression displayed increased disease-free survival (DFS) and overall survival (OS) than HCC patients with high miR-221 expression (long-rank test: *p* < 0.001). Furthermore, the authors showed that tumor size, AFP protein level, microvascular invasion, and miR-221 were significantly correlated with DFS and OS. Finally, a multivariate Cox analysis suggested that miR-221 was an independent indicator of poor prognosis in HCC patients.

Another indication of miR-221 involvement in HCC derives from the work performed by Yousurf et al. [78]. The group identified a three-miRNA panel, including miR-221, with improved diagnostic efficiency in differentiating HCC patients from healthy controls. The expression profile of miR-221 in the serum of 33 HCC patients and 33 controls was studied using real-time qRT-PCR and was found to be downregulated in HCC. In contrast, miR-221 expression levels in tumor specimens were significantly higher than in the peri-tumor counterparts. It would be interesting to further study the contrasting levels of miR221 in HCC tissue and the serum; unfortunately, the authors did not comment on this peculiar aspect. ROC analysis was used to validate the discriminatory potential of significantly altered miRNAs. Among nine differentially expressed serum miRNA in HCC patients, miR-221 showed the second-best diagnostic efficacy, with a rather high Area Under the Curve (AUC) value (AUC 0.786, 95% CI = 0.666–0.906, cut-off < 1.626; 228, sensitivity 77.14%, specificity 80.77%, *p* < 0.0001). The authors also observed that the diagnostic efficacy of miR-221 alone was superior to that of AFP. Importantly, it was highlighted that the best diagnostic efficacy occurred when miR-221 expression was associated with that of miR-221 and Let-7a. This observation reinforces the concept that evaluating the expression levels of a single miRNA may not be optimal for precise HCC diagnosis.

Finally, a recent meta-analysis [79] found that the overexpression of miR-221 was significantly associated with poor OS (HR  =  1.91, 95% CI: 1.53–2.38, *p* < 0.01) and DFS (HR  =  2.02, 95% CI: 1.58–2.57, *p* < 0.01) when the results of seven articles describing 416 patients and five studies with 391 patients were compared. A subgroup analysis of OS showed that high miR-221 expression in HCC tissue was closely associated with poor prognosis regardless of the ethnic group (Asian vs. non-Asian patients) or the specimen used for detection (formalin-fixed, paraffin-embedded tissue or frozen tissue).

#### 4.1.2. miR-487a

Recent studies have demonstrated that miR-487a plays a key role in cancer progression, including gastric and breast cancers. In addition, it targets the mRNA of TIA1 [80], a protein that binds the 3′ untranslated region (3′UTR) of mRNA and the mRNA of membrane-associated guanylate kinase inverted 2 (MAGI2) [81], a protein that seems to act as a scaffold molecule at synaptic junctions. Both TIA1 and MAGI2 are involved in the TGFβ-induced epithelial–mesenchymal transition.

Chang R-M. et al. [82] studied a total of 198 cases, which were randomly divided into two groups: a training cohort (132 patients) and a validation cohort (66 patients). In both groups, the authors demonstrated that miR-487a expression in HCC tissues was higher than that in adjacent non-tumorous liver tissues in 85.6% (113/132) and 81.8% (54/66) of cases in the training and validation cohorts, respectively. In addition, to determine whether the miR-487a level correlated with the prognosis of HCC patients, cases from the training and validation cohorts were divided into two additional subgroups consisting of patients whose miR-487a expression in HCC tissues was 2-fold higher than that in their adjacent non-tumorous liver tissues (the high-expression group) and the remaining samples (the low-expression group). In the training cohort, miR-487a levels were directly associated with tumor size, nodule number, capsule formation, microvascular invasion, and tumor node metastasis classification. In contrast, in the validation cohort, miR-487a expression was only associated with tumor size, nodule number, and microvascular invasion. To understand whether miR-487a could be a prognostic factor, OS and DFS were analyzed in both groups. The authors observed that OS and DFS were worse in the high miR-487a expression group compared to the low miR-487a expression group in both cohorts. Using univariate analysis, it was demonstrated that liver cirrhosis, the number of nodules, microvascular invasion, and miR-487a expression were independent risk factors for OS and DSF in the training cohorts. In the validation cohort, the number of nodules, microvascular invasion, tumor node metastasis classification, and miR-487a levels were independent risk factors for OS, whereas only microvascular invasion and miR-487a expression were independent risk factors for DFS. All this evidence suggests that the upregulation of miR-487a is an independent risk factor and predictor of poor survival for HCC patients.

#### 4.1.3. miR-33a

miR-33a is a member of the miR-33 family, which is highly conserved in human species. It is an intronic miRNA within the genes of sterol regulatory element-binding proteins (SREBPs) [83]; it primarily regulates the metabolism of cholesterol [84] and glucose [85]. The role of miR-33a in cancer has been described in the literature, with numerous studies indicating that it exerts tumor-suppressive effects in certain cancers, including pancreatic and lung cancers [86]. Another study has demonstrated its implication in the development of chemo-resistance in osteosarcoma [87]. Furthermore, miR-33a has been demonstrated to significantly increase in liver tissue in a fibrosis progression-dependent manner [88].

In 2018, Ru-Ting Xie et al. [89] analyzed 149 HCC biopsies, 36 of which were paired with para-carcinoma tissues. The authors found that patients with lower miR-33a expression had significantly poorer survival than those with higher miR-33a levels. miR-33a expression was significantly correlated with multiple foci; moreover, multivariate Cox analysis suggested that the combination of low miR-33a levels and multiple foci was associated with a significant decrease in OS and progression-free survival (PFS). Conversely, a recent paper reported that a reduced miR-33a level was associated with an increased survival probability [90]. Given the presence of conflicting data in the existing literature regarding its role and prognostic value, further investigation is required to unravel the significance of miR-33a.

#### 4.1.4. miR-105-1

miR-105-1 represents a member of the miR-105 family, which is located within the X chromosome [91]. miR-105-1 can affect Annexin A9 (ANXA9) expression, enhancing chemo-resistance and attenuating the cell apoptosis induced by cisplatin in cisplatin-sensitive ovarian cancer cells [92].

miR-105-1 has been previously identified as a key regulator of circulation, the pre-metastatic process, and metastatic progression in early breast cancer [93]. Additionally, low levels of miR-105-1 correlate with poor clinical outcomes and aggressive progression in glioma [94]. In 2014, the expression of miR-105-1 was observed to be markedly downregulated in both HCC cell lines and HCC tissues [95]. The authors also showed that miR-105-1 overexpression suppressed HCC cell proliferation in vitro and in vivo by targeting the FosfoInositide-3-Kinasi (PI3K)/AKT serine/threonine kinase 1 signaling pathway, thus suggesting a role as a tumor suppressor.

In 2017, Yu-Shui Ma et al. [96] confirmed the above observation, demonstrating, through in silico and real-time qRT-PCR analysis, reduced miR-105-1 expression in an HCC liver biopsy compared to non-tumor liver tissues. Furthermore, the authors showed that reduced expression was associated with HCC progression and poor prognosis. Via overexpression and silencing assays, it was demonstrated that nuclear receptor coactivator 1 (NCOA1) is a direct target of miR-105-1. NCOA1 is a transcriptional coregulatory protein recruited to DNA promoter sites, where it acylates histones, making downstream DNA more accessible to transcription. The authors observed that NCOA1 was upregulated in HCC tissue compared to normal liver and that its upregulation was related to a poorer OS rate. Thus, the reduced expression of miR-105-1 in HCC creates an increased level of NCOA1 with consequent pathological transcriptional activation.

#### 4.1.5. miR-138-5p

The human miR-138 family consists of miR-138-3p and miR-138-5p, located on chromosomes 3p21.32 and 16q13, respectively. Recent reports have demonstrated that miR-138-5p exerts a repressive effect on the proliferation of endothelial progenitor cells by inhibiting hypoxia-inducible factor-1α (HIF-1α) and vascular endothelial growth factor A (VEGFA), both master regulators of angiogenesis and tumor vascular mimicry [97]. In 2022, Liu et al. [98] demonstrated that miR-138-5p exhibited lower expression in HCC tissues compared to para-tumor tissues and that low levels of miR-138-5p were significantly associated with high vascular mimicry density and high levels of HIF-1α and VEGFA. Furthermore, the authors discovered a negative correlation between miR-138-5 expression and HIF-1α mRNA levels. A Kaplan–Meier analysis of survival performed on a dataset from “The Cancer Genome Atlas” (TCGA) revealed that low miR138-5p expression and high HIF-1α/VEGFA mRNA levels were associated with a poor prognosis in patients with HCC.

Furthermore, in 2022, Jiang’s research group [99] demonstrated that the non-coding circular RNA circ-TLK1 functions as an oncogene in HCC progression, partly through the inhibition (sponge effect) of miR-138-5p. Notably, the expression of miR-138-5p was increased in the HCC cell line HepG2 following the knockdown of circ-TLK1; in contrast, the overexpression of circ-TLK1 resulted in a reduction in miR-138-5p levels in the HCC cell line HuH7. These observations demonstrate the functional relationship between circ-TLK1 and miR-138-5p. A comparative analysis of circ-TLK1 expression profile in 87 specimens from HCC patients revealed significant upregulation of circ-TLK1 in HCC tissues compared to adjacent non-tumor tissues. Moreover, high intra-tumor circ-TLK1 expression appeared to be closely correlated with a larger tumor size (*p* = 0.030), an advanced TNM stage (*p* = 0.018), and vascular invasion (*p* = 0.005). Additionally, lower levels of miR-138-5p were observed in HCC tissues compared to normal tissue. Notably, miR-138-5p expression was negatively correlated with that of circ-TLK1 in HCC. This is the first demonstration that circ-TLK1 is highly expressed in HCC specimens and acts as an oncogene in HCC progression, partly through competitive binding to miR-138-5p. This, in turn, relieves the inhibitory effects of miR-138-5p on its target SRY-related HMG-box (SOX4). SOX4 belongs to a family of transcription factors whose overexpression favors early HCC recurrence [100].

In 2020, Xiao Xi J. et al. [101] investigated the impact of the miR-138/SOX4 axis on Anillin, an actin-binding protein that plays a pivotal role in regulating cytokinesis during the cell cycle and is, thus, involved in tumorigenesis progression. In vitro assays demonstrated that both SOX4 and Anillin mRNA expression were significantly reduced by miR-138 upregulation in HCC cell lines. Furthermore, the re-introduction of SOX4 into HCC cells increased Anillin levels, even in the presence of miR-138 upregulation. The authors investigated the expression profile of Anillin in cancer through an analysis of the TCGA database. They found that Anillin is commonly highly expressed in most solid human tumors, including HCC. In order to gain further insight into the expression profile of Anillin, the researchers conducted a detailed evaluation of its expression in 87 HCC tissues compared to their adjacent non-tumor tissues. In the tumor tissues, approximately 67.8% (59/87) of samples exhibited high expression of Anillin, whereas in non-tumor tissues, high expression of Anillin was observed only in a small portion of cases (25.3%; 22/87). The remaining cases displayed markedly low Anillin expression compared to the tumor tissue specimens. The DFS and the OS in a dataset of 361 HCC patients revealed a significant correlation between higher Anillin levels and poor prognosis, as well as high mortality. Furthermore, DFS and OS rates were markedly dismal in patients with high Anillin expression.

#### 4.1.6. miR-21

miR-21 is one of the most extensively researched oncogenic microRNA, with many studies conducted on its role in various cancers. It is located within the Vacuole Membrane Protein 1 (VMP1) locus on chromosome 17 [102].

miR-21 has been identified as a particularly significant factor in HCC. It is upregulated in tumor tissue, and its overexpression has been associated with poor OS in HCC patients [103]. In particular, paired tissue (HCC and normal liver tissue), serum, and serum exosome sequencing has indicated the positive correlation of miR-21 between serum exosomes and HCC tissue, suggesting that miR-21 is exported from HCC tissue to circulation via exosomes.

In 2023, Stechele et al. [104] investigated the potential value of circulating miR-21 as a therapy response marker for HCC patients treated with high-dose-rate brachytherapy. The researchers demonstrated that an increase in the plasma of miR-21 level two days after ablative therapy was associated with a poor response to therapy and a shorter time to systemic progression. These data are consistent with the results of previous studies demonstrating a correlation between high miR-21 expression in HCC tissue and poor prognosis [105]. In particular, miR-21 levels were measured in 166 samples of surgically resected HCC nodules. The investigation’s results indicated that high miR-21 expression represents an independent prognostic factor for both short OS (hazard ratio 2.36) and DFS (hazard ratio 2.02).

Another study [106] indicated that circulating miR-21 and miR-130b may serve as potential tumor biomarkers for diagnosing liver cancer. The combined detection of serum miR-130b and miR-21 was demonstrated to be an effective approach to the diagnosis of HCC, showing a sensitivity of 92.16% and an accuracy rate of 77.51%. The evidence was obtained through a comparative analysis of results obtained from serum samples and tumor tissues from 46 patients with HCC and 55 healthy volunteers. The levels of miR-21 and miR-130b were found to be upregulated in both serum levels and tumor tissues when compared to the corresponding control group. Spearman’s rank analysis revealed a highly significant positive correlation between serum and tumor tissue levels. Comparing the serum expression profile of miR-21 and miR-130b in HCC patients before and after surgical treatment, the authors observed a marked decrease in the expression level of both miRNAs two weeks after surgery compared to the levels observed before surgical treatment. Kaplan–Meier analysis indicated a significant difference in OS and PFS between the high-serum miR-130b/Mir-21 expression group and the low-expression group. Moreover, the authors identified a correlation between the circulating levels of both miR-21 and miR-130b and tumor capsular infiltration, HCC stage, and distant metastasis.

#### 4.1.7. miR-122

In humans, miR-122 is encoded at a single genomic locus on chromosome 18, and its transcription is regulated by hepatocyte nuclear factor 4 alpha (HNF4α) [107]. miR-122 is highly expressed in the liver, and it has been demonstrated to regulate a range of physiological and pathological conditions, including lipid metabolism, the response to drug administration/viral infection, alcoholic hepatic injury, and the formation of fibrosis [108]. Previous research has shown that miR-122 expression is downregulated in HCC tissues compared to normal liver tissues and that it functions as a tumor suppressor during the process of hepato-carcinogenesis [109].

In 2020, a meta-analysis was conducted to evaluate the diagnostic performance of serum miR-122 in HCC patients compared to a control group [110]. A comparison of data from 13 studies, which included 920 HCC patients and 1217 healthy controls, revealed that serum miR-122 exhibited moderate diagnostic accuracy in the discrimination of HCC (AUC = 0.82). Further, the analysis indicated that the serum miR-122 level was an acceptable discriminator of HCC patients from healthy individuals or those with HBV or HCV (AUC = 0.87) infection. However, discriminating between cirrhotic and HCC patients was less effective (AUC = 0.74). Taken together, the data indicated that the measurement of circulating miR-122 potentially offers moderate diagnostic efficacy for discriminating HCC patients from healthy individuals or those affected by HBV or HCV.

In 2022, Fang Y. et al. [111] conducted a case–control study, which included 100 healthy controls and 100 patients with early-stage HCC. The aim was to evaluate the diagnostic efficacy of a profile of differentially expressed miRNAs, including miR-122. The results demonstrated a markedly increased expression level of circulating miR-16 and miR-122 in the serum of early-stage HCC patients compared to the control group. More specifically, the AUC value of miR-16 was 0.798, with a sensitivity of 91% and a specificity of 58% (95% CI 0.738–0.858, *p*  <  0.001), whereas the AUC value of miR-122 was 0.759, with a sensitivity of 83% and a specificity of 64% (95% CI 0.690–0.827, *p*  <  0.001). The author also evaluated the diagnostic power of the combination of the two miRNAs compared to AFP. The AUC value for the combination of miR-16 and miR-122 increased to 0.803, although with a lower sensitivity (58%) but a higher specificity (84%) (95% CI 0.745–0.861, *p*  <  0.001). The AUC value of AFP was lower than that observed for the miRNA combination (0.716), with a sensitivity of 64% and a specificity of 84% (95% CI 0.641–0.791, *p*  <  0.001). The combination of AFP with either miR-16 or miR-122 not only increased the AUC value (to 0.859 and 0.825, respectively) but also the specificity (to 87% and 79%, respectively). Finally, the combination of AFP with both miRNAs yielded the highest AUC value (0.862), accompanied by a sensitivity of 76% and a specificity of 80% (95% CI 0.813–0.911, *p*  <  0.001). Thus, it can be concluded that these circulating miRNAs have significant value in the early screening and diagnosis of HCC when combined with AFP.

In 2021, an Italian clinical study [112] compared the levels of circulating miR-21 and miR-122 in healthy subjects and HCC patients. The aim was to evaluate the role of these miRNAs as predictors of PFS in a group of Caucasian HCC patients treated using drug-eluting bead trans-arterial chemoembolization (DEB-TACE). Serum miR-122 levels demonstrated a progressive increase in HCC patients compared to controls (*p* = 0.02). The increase was associated with the underlying etiology, particularly virus-related liver disease. ROC curves were used to identify optimal prognostic cut-offs for serum miR-21 and miR-221 levels, ensuring maximal sensitivity and specificity. In contrast to miR-21, miR-122 levels immediately before DEB-TACE treatment proved to be a predictive factor for PFS. Indeed, patients with miR-122 levels below the defined cut-off exhibited a significantly longer median PFS than that observed in the comparison group (5.6 months [1.4–9.7] vs. 5 months [1.8–3.2]). Notably, in a mouse model of HCC, a novel interconnection between miR-122 and hypoxia-induced pathways was identified [113]. In this regard, a correlation analysis was conducted between miR-21, miR-122, and HIF-1α in HCC patients undergoing DEB-TACE treatment. As a result of this investigation, a positive correlation between miR-21 and HIF-1α in HCC patients was observed both pre- and post-surgical treatment. In contrast, no significant associations with miR-122 were identified.

#### 4.1.8. miR 139-5p

miR-139-5p is located within the sequence of the phosphodiesterase 2A (PDE2A) gene [114]. The various recognized targets for this miRNA include Rho-associated coiled-coil kinase2 (ROCK2) [115]. ROCK2 promotes cell proliferation and migration in many tissues, including the liver, via interaction with multiple targets, such as cyclin D1 [116] and matrix metalloproteinase 2 (MMP-2) [117], while reducing the levels of p27Kip1 [116]. In HCC, miR-139-5p downregulation is related to the methylation of its promoter, a phenomenon occurring in hepato-carcinogenesis. For a decade, it has been known that miR-139-5p is downregulated in HCC and that its low expression relates to poor prognosis [118]. Additionally, a reduced expression level relates to vein invasion and the HCC stage. More recently [119], it has been observed that high tissue expression of miR-139-5p (hazard ratio: 0.72, *p* < 0.001) is associated with good prognosis in HCC patients. In [120] in vitro and in vivo models of HCC, we observed that the reactivation of miR-139-5p expression via demethylating drugs can reduce HCC cell proliferation and migration, thus contrasting HCC growth. Together, the above observations strengthen the rationale to consider miR-139-5p as a promising HCC diagnostic/prognostic marker and possibly as a therapeutic target [42].

**Table 1 ijms-25-12235-t001:** miRNAs considered as potential biomarkers for hepatocellular carcinoma.

miRNA	Prognostic Significance	Levels in HCC	References
miR-221	Poor	Increased (tissue) Decreased (serum)	[76,77,78,79]
miR-18a	Poor	Increased (tissue)	[76]
miR-487a	Poor	Increased (tissue)	[82]
miR-33a	Poor/Good	Reduced (tissue)	[89,90]
miR-105-1	Poor	Reduced (tissue)	[95,96]
miR-138-5p	Poor	Reduced (tissue)	[98,99,100]
miR-21	Poor	Increased (tissue/serum)	[103,104,105,106]
miR-122	Poor	Decreased (tissue, serum)	[110,111]
miR-139-5p	Poor	Decreased (tissue)	[118,119,120]

### 4.2. LncRNAs

LncRNAs have been associated with human pathologies in general and HCC in particular. For this type of molecule, less information about their biological behavior is known compared to miRNA. Despite this, they are actively investigated as potential diagnostic molecules in HCC. Among the many works published in the field so far, we focused on a selection that we believe is representative (Table 2).

#### 4.2.1. LncRNAs: HULC and MALATI

Initially identified as a highly upregulated transcript in the blood/cancer tissue of HCC patients [121], HULC is a 482 lncRNA encoded by a gene located on locus 6p24.3. Its aberrant expression has been associated with HCC development and progression [121]. HULC acts as a molecular sponge for miR-2001-3p, miR-186, and miR-107, thereby increasing the levels of multiple oncogenes, including ZEB1, HMGA2, and E2F1, respectively [122]. A recent meta-analysis published by Lumkul et al. [123] demonstrated that the serum levels of HULC, HOTARI, and UCA1 were increased in HCC patients. Moreover, they exhibited excellent discriminatory capabilities between HCC patients and patients with other forms of liver disease (AUC = 86%, 95% CI, 83–89%). Notably, the detection of serum HULC showed the best performance compared to HOTAIR or UCA1.

In 2018, Wang et al. [124] evaluated the role of several genetic polymorphisms of HULC in affecting cancer susceptibility and clinical outcomes. The researchers identified a specific polymorphism, rs1041279, that was associated with a 1.41-fold increased HCC risk in 521 patients when compared to 817 sex- and age-matched controls. However, no significant association was observed between HULC polymorphisms and OS. In the same years, Sonohara et al. [125] investigated the prognostic value of lncRNA HULC and MALAT1 (see below) following tumor surgical resection. Compared to the control group, the expression levels of HULC were found to be more elevated in HCC samples compared to normal tissue; MALAT1 expression did not differ between the control and cancerous tissues. A positive correlation was observed between the expression levels of MALAT1 and both capsular formation and tumor size. Both MALAT1 and HULC expression were found to be negatively correlated with the level of AFP. When patients were stratified by HULC levels, those with increased HULC expression tended to have improved relapse-free survival, although the difference was not statistically significant. Preliminary HULC analysis with Gene Expression Omnibus (GEO) and in silico analysis showed that increased HULC expression correlates with a better prognosis in HCC [126]. Altogether, these findings suggest that the increased expression levels of HULC in HCC tissue may serve as a promising prognostic biomarker for HCC.

MALAT1, initially described in non-small-cell lung cancer in 2003 [127], is an 8.5 kb lncRNA located on the 11q13 locus. It acts as a proto-oncogene involved in modulating several molecular signaling pathways related to cell proliferation, death, migration, invasion, immunity, angiogenesis, and tumorigenicity. A recent case–control study published by Golam et al. in 2024 [128] evaluated the diagnostic potential of MALAT1 in HCC compared to the routinely used diagnostic biomarkers. The MALAT1 blood levels were found to be significantly upregulated in both the HCV patient cohort and the HCC/HCV patient cohort in comparison to the healthy control group. Additionally, the MALAT1 expression level was increased in the HCV group compared to the HCC/HCV group. This observation may indicate that MALATI is more specific for HCV diagnoses than HCC detection. An ROC analysis was performed to assess the diagnostic efficacy and identify the optimal cut-off value for MALAT1 and the routinely employed biomarker AFP. Unfortunately, MALAT1 exhibited lower accuracy and specificity than AFP.

In 2023, Liao et al. [129] published a study based on multi-omics analysis and real-time qRT-PCR validation to investigate the prognostic value of MALAT1. A total of 368 patients with HCC were included in the analysis based on data from The Cancer Genome Atlas (TCGA) liver hepatocellular carcinoma (TCGA-LIHC) database. The authors identified five protein-coding genes (AGO2, HNRNPC, EZH2, SFPQ, and SRSF1) associated with worse survival when highly expressed. In addition, the authors showed that MALATI is overexpressed in HCC tissue compared to non-tumor tissue. It is, thus, possible that MALATI contributes to HCC by interacting with AGO2, HNRNPC, EZH2, SFPQ, and SRSF1. In Kaplan–Meier analysis, patients with increased MALAT1 expression showed shorter PFS (*p* = 0.033) and OS (*p* = 0.023) than patients with low expression. MALAT1 expression not only positively correlated with more than 20 mutations in genes related to HCC progression but also with epigenetic modification. In particular, the authors found that MALAT1-associated methylation was related to HCC patients’ poor survival. This suggests that MALAT1 may be involved in the progression and prognosis of HCC through its DNA-methylation effect.

#### 4.2.2. LncRNAs: Linc00152

In 2015, Linc00152, an 828 bp lncRNA located at chromosome 2p11.2, was first identified as a potential marker for HCC diagnosis [130]. In this study, conducted by Li et al., it was observed that the levels of Linc00152 and HULC were significantly upregulated in plasma samples obtained from HCC patients who had undergone hepatic resection compared to those obtained from healthy controls. Furthermore, the levels of Linc00152 were also significantly correlated with differentiation grade, tumor size, and HCC stage. The ROC curve analysis of the two lncRNAs demonstrated a substantial diagnostic value of HULC and Linc00152 for HCC diagnosis, with AUC values of 0.78 and 0.85, respectively. The combination of HULC and Linc00152 resulted in an increased area under the ROC curve value up to 0.87, thereby demonstrating the highest ability for discrimination between HCC patients and controls. Subsequently, in vitro studies were conducted to further elucidate the effects of Linc00152, demonstrating that this lncRNA plays a pivotal role in regulating cell proliferation, clonogenicity, and migration [131].

In 2020, Wang B. and colleagues [132] investigated the potential diagnostic role of lncRNA NRADI and Linc00152 in HCC. The study cohort comprised 63 patients with primary HCC who had undergone surgical resection. The authors observed that NRAD1 was expressed in 47.6% (30 of 63) of HCC patients, whereas Linc00152 was expressed in 71.4% (45 of 63) of HCC patients. Furthermore, the levels of the two lncRNA were significantly higher in HCC samples than in non-tumor tissues. The overexpression of NRAD1 and LincC00152 was found to be associated with a reduction in both OS and PFS rates. The univariable analysis demonstrated that microvascular invasion, tumor size, and high expression levels of NRAD1 and Linc00152 were significantly associated with the prognosis of patients with HCC. No data were provided regarding the plasma or serum levels of these two lncRNAs. Taken together, these data suggest that the overexpression of lncRNA NRAD1 and LINC00152 represents an independent risk factor associated with the prognosis of HCC patients.

In 2023, Shehab-Eldeen et al. [133] demonstrated that the serum expression levels of Linc00152 and UCA1 were significantly higher in patients with HCC than those with liver cirrhosis or healthy subjects. The ROC curve analysis indicated that Linc00152 and UCA1 exhibited high sensitivity (81.7% and 83.3%, respectively) and specificity (63.3% and 83.3%, respectively) for HCC, thereby establishing their efficacy as diagnostic biomarkers. Moreover, combining the two lncRNAs with AFP resulted in a significant increase in the accuracy of HCC diagnosis, with an AUC area of 0.99, a sensitivity of 100%, and a specificity of 95%. These findings collectively indicate the potential utility of serum Linc00152 and UCA1 as non-invasive biomarkers for HCC.

#### 4.2.3. lncRNA HOTAIR

HOTAIR (homeobox transcript antisense intergenic RNA) is located in the Homeobox C gene cluster, which is a spliced and poly-adenylated RNA with 2158 nucleotides and six exons. It originates from the transcription of the antisense strand of the HoxC gene, situated on chromosome 12q13.13 [134]. HOTAIR has been demonstrated to exhibit excellent discriminatory capabilities between patients with HCC and those with other forms of liver disease. Additionally, it is involved in tumorigenesis, metastasis, and drug resistance in various cancer types, including HCC. Ishibashi et al. [135] published one of the earliest studies investigating the clinical significance of HOTAIR expression in HCC in 2013. By staging patients according to their HOTAIR expression levels in HCC tissue, the authors demonstrated that patients with high HOTAIR expression exhibited a markedly poorer prognosis, reduced OS, and a significantly larger tumor size than those with low HOTAIR expression. In a study published in 2021, Han and colleagues [136] reached similar conclusions regarding the role of HOTAIR in advanced HCC and its correlation with the use of Sunitinib as a chemotherapy agent. A total of 60 patients with a diagnosis of advanced HCC who had not received any form of treatment before Sunitinib treatment were enrolled in the study. The authors demonstrated that the expression of HOTAIR was significantly higher in tumor tissues than in adjacent non-tumor tissues. Furthermore, HOTAIR levels in the peripheral blood of HCC patients were higher than those observed in healthy subjects. Moreover, a positive correlation was identified between HOTAIR levels in tumor tissue and in the peripheral blood of HCC patients. From a prognostic perspective, patients with low levels of HOTAIR in tumor tissues displayed significantly prolonged survival (13.4 vs. 9.5 months) and PFS (8.4 vs. 6.2 months) compared to those with high expression levels. Similarly, patients with low HOTAIR levels in peripheral blood demonstrated significantly higher OS (12.8 vs. 9.1 months) and PFS (8.9 vs. 6.4 months) than those with high expression. Patients with low expression in both tumor tissue and peripheral blood exhibited the most promising OS and PFS outcomes, with median OS and PFS times of 14.3 and 10.6 months, respectively, compared to the remaining patient cohort. The results of COX regression analysis indicated that the levels of HOTAIR in tumor tissue and peripheral blood were independent predictive factors of OS and PFS in patients with advanced HCC who were treated with Sunitinib.

**Table 2 ijms-25-12235-t002:** lncRNAs considered as potential biomarkers for hepatocellular carcinoma.

lncRNA	Prognostic Significance	Levels in HCC	References
HULC	Poor	Increased (serum, tissue)	[121,123,124,125,130]
MALAT1	Poor	Increased (serum/tissue)	[128,129]
Linc00152	Poor	Increased (plasma, tissue)	[130,132,133]
HOTAIR	Poor	Increased (tissue, serum)	[135,136,137,138]

In 2022, a further study conducted by Lou et al. [137] provided additional evidence to support the potential role of serum HOTAIR as a marker for early diagnosis and prognosis in HCC patients. A total of 61 patients diagnosed with HCC and a control group comprising both patients with liver cirrhosis and healthy subjects were included. Serum levels of HOTAIR, as well as BRM and ICR, were all found to be significantly increased in patients with tumors in comparison to both patients with liver cirrhosis and healthy controls. Serum HOTAIR levels were closely associated with the HCC stage, metastasis, vascular invasion, and tumor size. A notable reduction in the serum levels of HOTAIR and ICR was detected between the assessments conducted during the preoperative visit and one week after tumor resection. The ROC curve analysis demonstrated that the three lncRNAs alone showed higher accuracy for identifying HCC compared to AFP. Notably, HOTAIR showed the highest AUC (0.991 with 96.7% sensitivity and 95% specificity) in distinguishing HCC patients from healthy controls. The COX regression analysis demonstrated that the serum levels of HOTAIR, BRM, and ICR were significantly correlated with HCC prognosis. In conclusion, the data collectively indicate that the serum levels of these three lncRNAs could serve as a panel of markers for early HCC diagnosis and possibly for monitoring therapeutic effects.

A recent study published in 2024 further investigated the role of HOTARI as a potential biomarker for HCC. The meta-analysis by Wen et al. [138] included eight studies comprising 399 subjects. The statistical analysis revealed a significant correlation between high HOTAIR levels and an advanced HCC stage (OR = 1.45; 95% CI [1.02, 2.05]; *p* = 0.038), distant metastasis (OR = 2.51; 95% CI [1.02, 6.14]; *p* = 0.044), and poor tumor differentiation (OR = 1.59; 95% CI [1.21, 2.10]; *p* = 0.001). No correlations were found between HOTARI expression levels and age, gender, tumor number, tumor size, lymph node metastasis, or AFP level. Among the eight studies, only three identified a negative correlation between OS and HOTARI expression levels. Patients were classified into two groups based on their expression levels of HOTARI. The findings revealed that patients in the high-level HOTAIR group exhibited a poor OS rate (HR = 1.69; 95% CI [1.14, 2.51]; *p* = 0.009) compared to the low-level HOTAIR group. Furthermore, a significant negative association was observed between HOTAIR expression and RFS (HR  =  1.89; 95% CI [1.37, 2.59]; *p* = 0.0001). In conclusion, the data suggest that HOTAIR may be a promising biomarker for evaluating HCC prognosis.

### 4.3. CircRNAs

CircRNAs are the last class of ncRNA described in this review as being associated with HCC and its diagnosis. Like lncRNAs, we also need to acquire more knowledge about this class of molecules’ biology. Despite this, their potential as HCC biomarkers is under active investigation. Among the many works published in the field so far, in the present review, we focus on some representative examples (Table 3).

#### 4.3.1. circ_0001445/cSMARCA5

Despite an increasing number of studies in recent years that have demonstrated the pivotal role of circRNA in regulating cellular functions and metabolism in cancer [139], only a limited number of circRNAs have a significant value in terms of HCC diagnosis and prognosis. One of the circRNAs correlated with poor prognosis is circ_0001445 (also named cSMARCA5). It is derived from exons 15 and 16 of the SWI/SNF-related, matrix-associated, actin-dependent regulator of chromatin, subfamily a, member 5 (SMARCA5) gene. Circ_0001445 acts as a sponge for miR-17-3p and miR-181b-5p, which regulate the levels of the tissue inhibitor of metalloproteinase-3 (TIMP3) implicated in matrix remodeling. In 2018, Zhang et al. [140] analyzed circ_0001445 levels in 73 pairs of HCC and adjacent non-tumor tissues, observing that circ_0001445 expression was significantly lower in tumor tissues than in adjacent non-tumor tissues. Furthermore, its expression was inversely correlated with the number of HCC foci. In HCC cells cultured in vitro, overexpression of circ_0001445 promoted apoptosis and inhibited proliferation, migration, and invasion, indicating a tumor-suppressor effect. The authors also analyzed circ_0001445 levels in the plasma of HCC (104), cirrhosis (57), and HBV (44) patients and healthy subjects (52). It was observed that circ_0001445 levels were markedly reduced in HCC patients in comparison to healthy controls, cirrhosis patients, or HBV patients. Notably, the levels of circ_0001445 in HCC patients were significantly lower than in cirrhotic and HBV patients. These findings suggest that circ_0001445 may be a useful tool to discriminate HCC patients from cirrhotic/HBV patients. ROC curve analysis was performed to assess the diagnostic value of circ_0001445 in discriminating HCC. It is noteworthy that circ-RNA circ_0001445 was a more effective diagnostic marker compared to AFP.

In 2020, Xu et al. [141] corroborated the previously reported results of circ_0001445 downregulation. The authors showed that its overexpression impairs HCC progression by downregulating the epithelial–mesenchymal transition (EMT) process and glycolysis in HCC cells in vitro. This phenotypic effect was due to the downregulation of the miR-942-5p/ALX4 (Aristaless-like homeobox 4) axis. In particular, circ_0001445 promoted ALX4 expression by targeting miR-942-5p. Notably, ALX4 is a transcription factor known to have anti-HCC effects [142].

In 2020, Wang et al. [143] investigated the correlation between multiple circRNA, including circ_0001445, and the clinical–pathological characteristics of HCC patients. The authors demonstrated that reduced blood levels of circ_0001445 correlated with increased HCC malignancy, as indicated by satellite nodules and frequent vascular invasion. These findings fully support the concept of the prognostic value of this circRNA.

In contrast to the above results, a recent case–control study [144] indicated that circ_0001445 exhibited higher expression in the specimens of 106 patients with HCC compared to adjacent non-tumor tissue. Furthermore, the authors demonstrated that high expression of circ_0001445 in HCC patients represented a risk factor for 3-year OS (hazard ratio = 1.798, 95% CI: 1.165~3.231, *p* = 0.0321). The results of the multivariate Cox analysis revealed that elevated circ_0001445 expression levels were associated with the development of portal vein tumor thrombus (PVTT), tumor size exceeding 5 cm, poor histopathologic grading, and multiple tumor foci. These discrepant results indicate the necessity to clarify the role of circ_0001445 as a biomarker for HCC.

#### 4.3.2. circ_0001649

circ_0001649 is a transcript product of the SHPRH (SNF2 histone linker PHD RING helicase) gene located on chromosome 6 [145]. SHPRH is a ubiquitously expressed protein that contains motifs characteristics of several DNA repair proteins, transcription factors, and helicases. Qin et al. [146] published one of the first studies on this circRNA in 2016. The authors evaluated the expression of circ_0001649 in 89 HCC samples and their paired adjacent liver tissues using qRT-PCR, showing significant downregulation in HCC tissue. They also observed that circ_0001649 expression was negatively associated with tumor emboli formation and tumor size. Moreover, in vitro silencing in HCC cells showed that the mRNA levels of MMPs (matrix metalloproteinases) 9, 10, and 13 were significantly increased after circ_0001649 knockdown. Notably, both the presence of tumor emboli and high levels of MMPs were correlated strongly with the invasive and metastatic properties of HCC. ROC analysis was performed to evaluate the prognostic power of circ_0001649. The AUC of circ_0001649 was 0.63, with a sensitivity and specificity of 81% and 69%, respectively. These data suggest that circ_0001649 has the potential to be used as a novel biomarker for HCC.

In 2018, Zhang et al. [147] confirmed the potential biomarker role of circ_0001649 in HCC by analyzing the correlation between its expression level and HCC prognosis. This study was conducted on 77 tumor samples and their paired adjacent liver tissues. The results of the survival analysis indicated that patients with low expression levels of circ_0001649 exhibited the poorest OS rates. The univariate analysis revealed that circ_0001649, HCC stage, and AST (transaminase) had a statistically significant impact on the OS of HCC patients. Furthermore, the multivariate analysis demonstrated that the expression levels of circ_0001649 and the HCC stage were independent prognostic factors for OS in HCC patients. Notably, the multivariate analysis also indicated that circ_0001649 levels were an independent prognostic factor for HCC patients.

#### 4.3.3. circ_0003570

circ_0003570 is a highly conserved transcript product belonging to the family of the sequence similarity 53member B (FAM53B) gene, located on chromosome 10. In 2018, Fu et al. [148] demonstrated that circ_0003570 is downregulated in HCC cell lines and tissues. The authors analyzed 107 pairs of tumor tissues and adjacent non-tumor tissues obtained from patients with HCC. Firstly, they evaluated the expression level of circ_0003570 in cancerous tissues and in HCC cell lines, observing that cancerous tissues, as well as tumor cell lines, exhibited a lower expression level of circ_0003570 compared to adjacent non-tumor tissues and the human non-tumorigenic hepatic cell line, respectively. Moreover, the authors observed a progressive and statistically significant decrease in the expression levels of circ_0003570 from chronic hepatitis (14 cases) to fibrosis (63 cases) and HCC. This suggests that the expression levels of this circRNA may undergo dynamic changes in response to different liver diseases. Finally, the researchers investigated the correlation between circ_0003570 levels and HCC’s clinical pathological features. Significant correlations were observed between cicr_0003570 levels and tumor size, differentiation, microvascular invasion, HCC stage, and serum AFP levels. The ROC curve analysis was performed to assess the potential diagnostic effectiveness of this circRNA, demonstrating an AUC of 0.70 with a sensitivity of 45% and a specificity of 86.8%. When the analysis was restricted to cirrhotic patients, the sensitivity increased to 69.7%, while the AUC and specificity remained comparable to those observed in the HCC patient group. These findings collectively indicate that circ_0003570 is significantly associated with the clinical and pathological characteristics of HCC patients and, thus, may serve as a potential biomarker for differentiating between various liver pathologies.

In 2022, Zhang et al. [149] demonstrated that circ_0003570 downregulates the growth, migration, and invasion of HCC cells; in a xenograft mouse model of HCC, the authors also observed that circ_0003570 overexpression results in a lower tumor volume/weight compared to controls. At the molecular level, it was shown that circ_0003570 binds miR-182-5p, thus regulating the repression of the downstream target gene StAR-related lipid transfer domain protein 13 (STARD13). STARD13 is implicated in HCC as it is downregulated in the less-differentiated HCC forms.

The study conducted by Jang et al. [150] further corroborated the notion of circ_0003570 downregulation in HCC. The clinical study included 162 HCC patients for which both tumor and adjacent non-tumor tissues were collected. The authors confirmed the reduced expression of circ_0003570 in HCC tissues vs. paired non-cancerous tissues. Subsequently, patients were stratified according to the levels of circ_0003570 and classified as high- or low-expression groups. Low circ_003570 was positively associated with a larger tumor size (>5 cm), vessel invasion, an advanced HCC stage, and a higher AFP serum level. The survival analysis revealed a significant association between the OS of HCC patients and the expression levels of circ_0003570. In particular, the cumulative 1-, 2-, and 4-year OS rates in the low-expression group were 40.0%, 31.1%, and 24.4%, respectively. In contrast, the OS rates increased to 68.1%, 62.7%, and 40%, respectively, in the high-expression group. The univariate analysis demonstrated that the significant predictors of OS were high circ_0003570 levels, multiple tumors, an AFP level higher than 200 ng/mL, chronic hepatitis B, and curative treatment. The multivariate analysis identified high circ_0003570 levels and curative treatment as independent prognostic factors for OS. Similarly, a significant difference in PFS was observed between patients stratified according to their circ_0003570 expression levels. The cumulative 1-, 2-, and 4-year PFS rates in the low-expression group were 31.1%, 22.2%, and 15.6%, respectively. In contrast, the PFS rates increased to 60.5%, 43.4%, and 26.3%, respectively, in the high-expression group. The univariate analysis demonstrated that the prognostic factors for PFS were a high circ_0003570 level, multiple tumors, an AFP level higher than 200 ng/mL, and curative treatment. The multivariate analysis identified independent prognostic factors for PFS as high circ_0003570 expression and curative treatment. Together, these data support a significant correlation between the circ_0003570 expression level with the clinical and pathological characteristics of HCC and the survival and progression of HCC patients.

Kang et al. further substantiated the prognostic value of circ_0003570 with regard to OS and PFS in HCC patients in 2023 [151]. The authors enrolled 86 patients with HVB-related HCC, who were divided into high- and low-expression groups based on the median expression levels of the circRNA. The Kaplan–Meier survival analysis revealed that patients with low circ_0003570 expression displayed significantly reduced OS. The cumulative 1-, 3-, and 5-year OS rates were 76.7%, 58.1%, and 51.2%, respectively, for patients with high expression levels, compared with 46.5%, 30.2%, and 30.2% for those with low expression levels. The multivariate analysis identified an advanced HCC stage, the presence of nodal involvement, metastasis, and sarcopenia as significant risk factors for OS in patients with HBV-related HCC. Similarly, the PFS rate was significantly affected in patients with low expression levels compared to those with high expression levels. The cumulative 1-, 3-, and 5-year PFS rates were 39%, 21.8%, and 21.8%, respectively, for patients with low expression levels, compared to 71.4%, 34.3%, and 21.8% for those with high expression levels. The multivariate analysis demonstrated that an advanced HCC stage and decompensated liver cirrhosis were significant risk factors for PSF in patients with HBV-HCC. Collectively, these data indicate that circ_0003570 expression levels are associated with favorable clinical and pathological features, as well as improved OS and PFS, in patients with HBV-HCC. It remains to be clarified whether this observation also holds for HCC induced by predisposing conditions other than HBV.

#### 4.3.4. Other circRNAs

In 2019, circ_0028502 and circ_0076251 were identified as potential biomarkers for diagnosis and OS in HCC patients [152]. Circ_0028502 is transcribed from Solute Carrier Family 8-member B1 (SLC24A6) located on chromosome 12, while circ_0076251 is transcribed from Zinc Finger AN1-type containing 3 (ZFAND3) located on chromosome 6. The study was conducted in 100 paired samples of HCC and adjacent liver tissues obtained from patients who had undergone surgical treatment. The authors demonstrated that both circRNAs were downregulated in HCC tissues compared to normal tissues. Furthermore, they observed a correlation between the expression level of circ_0028502 and the HCC stage; the levels of circ_0076251 were instead related to the presence of comorbidities, specifically type 2 diabetes mellitus and the presence of hepatitis B surface antigen (HbsAg). By extending the study population to include patients with chronic hepatitis (14) and liver fibrosis (56), the authors observed that both circRNAs were differentially expressed in relation to liver disease, with a progressive and significant decrease from chronic hepatitis to fibrosis and HCC. To evaluate the diagnostic efficacy of circ_0028502 and circ_0076251, ROC curve analyses were performed. The AUC for discriminating HCC from liver fibrosis and chronic hepatitis for circ_0028502 was 0.675, with a specificity of 72.1% and a sensitivity of 58%. Circ_0076251 demonstrated a higher AUC value (0.738) and sensitivity (64%) compared to circ_0028502 while maintaining a comparable specificity (71.3%). The combined diagnostic value of the two circRNAs was also evaluated, demonstrating a higher efficiency (AUC = 0.754) compared to using the two circRNAs independently. The Kaplan–Meier curve revealed that the survival rate differed depending on the expression levels of circ_0076251. Specifically, patients with low circ_0076251 levels exhibited a poorer OS rate than those with high expression levels.

CircDLC1 derives from exons 14, 15, and 16 of the DLC1 gene. In 2021, Liu et al. [153] demonstrated that circDLC1 was markedly downregulated in HCC tissues and that this was significantly correlated with prognosis. The Kaplan–Meier survival curves revealed a positive correlation between low circDLC1 levels and poor OS and RFS. In vitro and in vivo assays demonstrated that circDLC1 inhibits metastasis in hepatoma cells by interacting with the RNA-binding protein HuR and reducing the stability of matrix metalloproteinases 1 (MMP1) mRNAs.

Another circRNA identified as a potential biomarker for HCC progression is circ_0066659. Circ_0066659 is a 1254bp length RNA derived from the transmembrane protein 45A (TMEM45A) gene on chromosome 3. In 2020, Zhang et al. [154] investigated circRNA profiling in HCC samples and cell lines and selected circ_0066659 among the circRNAs found to be significantly upregulated. The study included paired HCC tissues and adjacent normal tissues obtained from patients with HCC. ROC curve analysis was employed to access the diagnostic value of circ_0066659, demonstrating high accuracy (AUC = 0.888). Furthermore, among patients exhibiting disparate circ_0066659 expression levels, HCC patients with low circ_0066659 expression displayed prolonged OS rate compared to those with high circ_0066659 expression, as evidenced by means of Kaplan–Meier survival curve analysis. Moreover, high expression levels of circ_0066659 were significantly correlated with tumor size, HCC stage, and vascular invasion. Via in vitro analysis, the authors confirmed the oncogenic role of cicr_0066659, showing its ability to promote cell growth. The authors demonstrated that Circ_0066659 mechanically sponges miR-665 expression, resulting in insulin growth factor 2 (IGF2) upregulation, whose overexpression accelerates the formation of liver tumors and HCC progression in mice [155].

**Table 3 ijms-25-12235-t003:** circRNAs considered as potential biomarkers for hepatocellular carcinoma.

circRNA	Prognostic Significance	Levels in HCC	References
0001445	Poor	Decreased (tissue, plasma) Increased (tissue)	[140,141,143,144]
0001649	Poor	Decreased (tissue)	[146,147]
0003570	Poor	Decreased (tissue)	[148,149,150]
0028502/0076251	Poor	Decreased (tissue)	[152]
CircDLC1	Poor	Decreased (tissue)	[153]
0066659	Poor	Increased (tissue)	[154]
0128298	Poor	Increased (tissue)	[156]

In 2018, Chen et al. identified circ_0128298 as a potential biomarker in the diagnosis and prognosis of HCC [156]. The study, which considered 78 HCC and para-tumor tissues collected from patients with HCC who underwent surgery, demonstrated that circ_0128298 is significantly upregulated in HCC tissues compared to adjacent non-tumor tissues. Increased levels of circ_0128298 were associated with the presence of vascular cancer embolus, lymphatic metastasis, and organ metastasis. The ROC curve was employed to investigate the diagnostic value of circ_0128298 in differentiating between HCC tissues and adjacent non-tumor tissues, demonstrating moderate accuracy (AUC = 0.668) and sensitivity (67.4%) but high specificity (80.5%). The univariate analysis demonstrated that gender, AFP levels, intrahepatic metastasis, organ metastasis, and circ_0128298 expression levels were independent factors in HCC. Furthermore, the multivariate COX regression analysis indicated that AFP levels and circ_0128298 expression levels were prognostic factors predicting poor survival among HCC patients. Finally, HCC patients who exhibited lower expression levels of circ_0128298 demonstrated prolonged OS compared to patients with higher levels of circ_0128298. These findings collectively suggest that circ_0128298 may be a predictive biomarker for HCC occurrence and prognosis. However, further research is required to confirm the diagnostic efficacy of circ_0128298 and gain a deeper understanding of the molecular and biological effects of circ_0128298 in HCC.

## 5. Conclusions

The increasing incidence of liver cancer, together with the poor effectiveness of treatments, especially for the advanced forms of the disease, makes early diagnosis/effective screening of the disease urgent. Instrumental diagnostic tools, despite being of great utility, can hardly be used as screening methods. In this context, liquid biopsy has arisen as a potential novel approach for HCC diagnosis. The identification of blood markers has many amenable features, including the absence of any risk for patients, the possibility of being used as a screening approach, and the possibility of performing multiple tests, allowing for the real-time monitoring of HCC evolution. Unfortunately, the available blood marker for HCC, i.e., AFP, has several limitations, as discussed at the beginning of the present review. In this context, employing ncRNAs may represent an interesting and novel diagnostic approach. Here, we present some noticeable examples of the many works published in the field so far. Altogether, these works strongly support the potential effectiveness of ncRNAs as diagnostic/prognostic markers for HCC. An evident advantage over AFP level determination is that for ncRNA detection, we can use real-time qRT-PCR and/or digital droplet PCR. These techniques are significantly more sensitive than those that detect AFP. Thus, it is reasonable to predict that ncRNA detection may be more precocious than AFP, with obvious advantages.

Despite this, some considerations must be made regarding using ncRNAs as HCC markers. First, it is unclear whether it is more appropriate to quantify ncRNAs directly isolated from plasma or extracellular vesicles (EVs) obtained from plasma. In EVs, ncRNAs are protected from blood nuclease degradation; thus, their presence and abundance may be enriched over those of ncRNAs freely circulating in the blood. Furthermore, it is not clear which ncRNAs circulate as naked molecules in the blood. Notably, none of the papers mentioned in this review specifically addressed this aspect. The second aspect concerns the relation between circulating ncRNAs and HCC tissue ncRNAs. In many of the studies mentioned, the authors limited ncRNA quantification to HCC tissue without checking blood levels. It is generally presumed that variations in the tissue levels are reflected in the amount in blood. However, this is not necessarily the rule; for example, Yousurf et al. [78] reported that miR-221 in the serum was downregulated, while in HCC tissue, it was upregulated. Third, due to the nature of miRNAs and perhaps that of lncRNAs/circRNAs being very promiscuous—as multiple ncRNAs regulate the same mRNA and an mRNA is regulated by many ncRNAs—it is possible that a signature of multiple ncRNAs is a more appropriate identifier, instead of a single ncRNA. In this regard, it is now well known that different types of tumors are associated with the dysregulation of the same ncRNA. Thus, for the specific identification of a given tumor, the quantification of multiple ncRNAs may be required. Fourth, to be considered a valuable marker of HCC, there must be concordant data in the literature about a given ncRNA. However, this is not always the case; for example, circ_0001445 was downregulated in HCC in some studies [140,141,142,143] and upregulated in another [144] (see Section 4.3, Circ-non-coding RNAs/Circ_0001445/cSMARCA5).

In conclusion, despite further studies being necessary to better characterize the biology of ncRNA in general and in relation to HCC, we believe that ncRNAs may represent attractive and novel markers for the early diagnosis and monitoring of HCC.

## Figures and Tables

**Figure 1 ijms-25-12235-f001:**
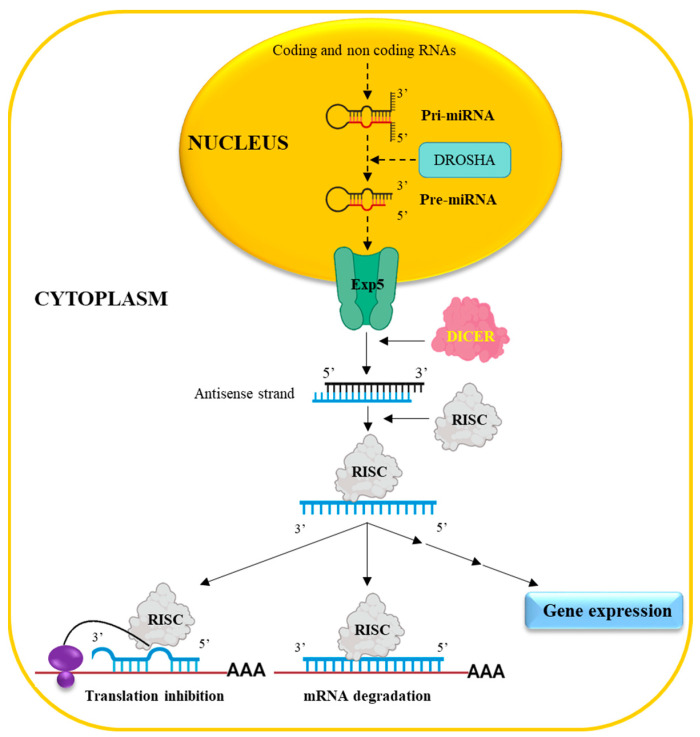
miRNA biogenesis and functions. In the cell nucleus, a long precursor named pri-miRNA is processed by the enzyme Drosha to pre-miRNA; this, in turn, is exported to the cytoplasm by the Exportin 5 enzyme. Here, the DICER enzyme produces double-stranded mature RNA (miRNA). The miRNA antisense strand is then loaded onto RISC, allowing for the recognition of the target mRNA, resulting in translation inhibition (via an imperfect base pairing) or mRNA degradation (via perfect base pairing). Recent findings indicate that via direct/indirect mechanisms, miRNA can also promote gene expression. This figure was created with BioRender.com (accessed on 23 September 2024).

**Figure 2 ijms-25-12235-f002:**
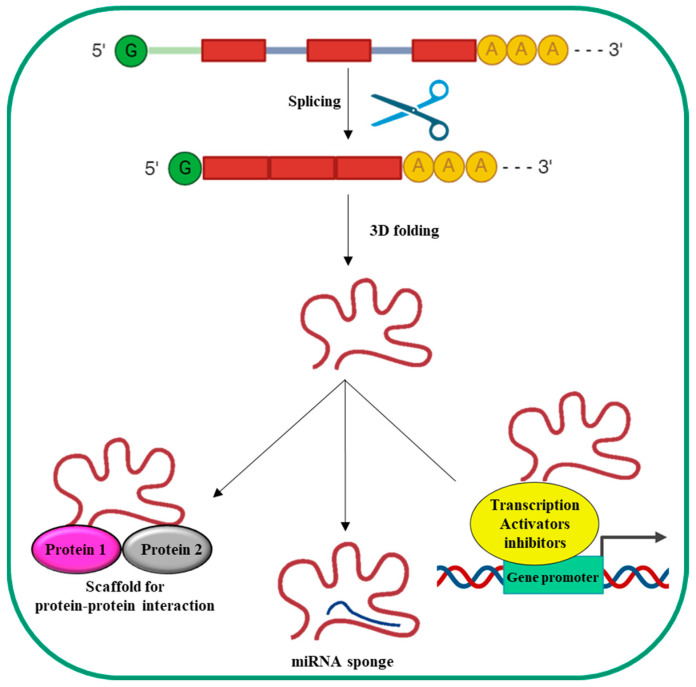
LncRNA biogenesis and functions. LncRNAs can be spliced, capped, and poly-adenylated. Following transcription, they assume a three-dimensional (3D) structure responsible for the biological effects, such as the recruitment of transcription activators/repressors to the promoters of their target genes, thus regulating gene expression; the inhibition of miRNA activity via the sponging effect; and the ability to act as scaffolds for protein to support the formation of protein complexes. This figure was created with BioRender.com (accessed on 23 September 2024).

**Figure 3 ijms-25-12235-f003:**
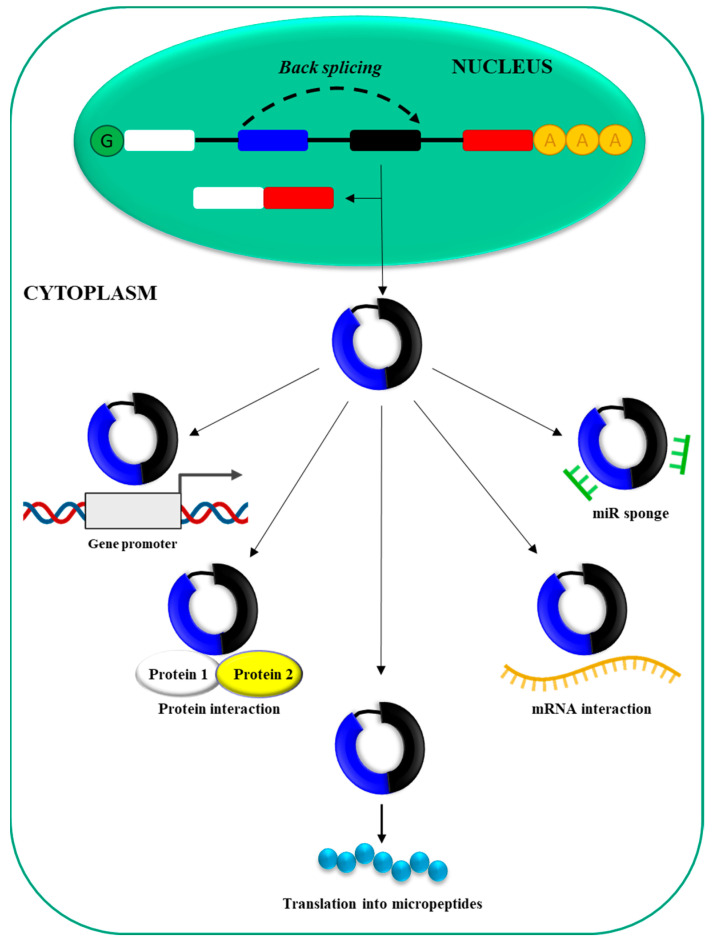
CircRNA biogenesis and functions. Although circRNAs have multiple biogenesis mechanisms, a common one is represented by back-splicing. Back-splicing can be induced by protein-dimerization, sequence complementarity of flanking introns, and exon-skipping mechanisms. Following the formation of a circular RNA, circRNA is exported into the cytoplasm, where it can bind miRNAs via complementary regions; undergo translation to generate small peptides; and interact with proteins, gene promoters, and specific mRNAs. This figure was created with BioRender.com (accessed on 23 September 2024).

**Figure 4 ijms-25-12235-f004:**
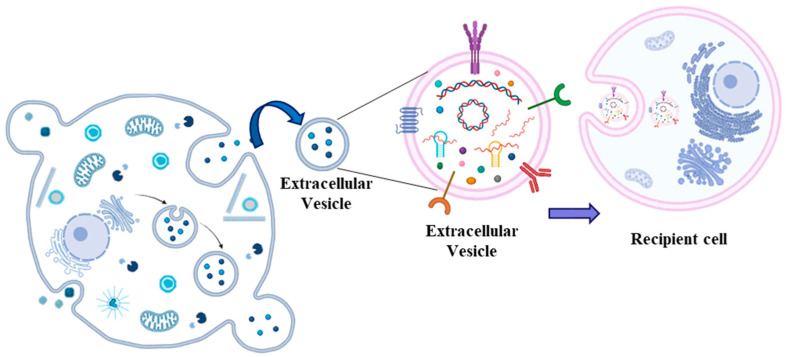
Extracellular vesicles. Extracellular vesicles (EVs) are a heterogeneous group of lipid bilayer particles synthesized and secreted by different cell types into the extracellular environment. EVs encapsulate various bioactive molecules, such as proteins, lipids, and nucleic acids. Nowadays, it is known that EVs can contain ncRNAs and can deliver these molecules to distant cells both under physiological and pathological conditions. This figure was created with BioRender.com (accessed on 23 September 2024).

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
