# Peer review of "Non-Coding RNAs as Potential Diagnostic/Prognostic Markers for Hepatocellular Carcinoma"

_ijms, 2024, doi:10.3390/ijms252212235_

Round 1

Reviewer 1 Report

Comments and Suggestions for Authors

While reviewing the manuscript titled "Non-coding RNAs as potential diagnostic/prognostic markers for hepatocellular carcinoma", I found that the authors:

- wrote a very good descriptive review.

- used illustrative diagrams to clarify some information.

- designed the review structure with good organization.

- used updated related references.

Author Response

While reviewing the manuscript titled "Non-coding RNAs as potential diagnostic/prognostic markers for hepatocellular carcinoma", I found that the authors:

- wrote a very good descriptive review.

- used illustrative diagrams to clarify some information.

- designed the review structure with good organization.

- used updated related references.

We wish to thank the reviewer for the appreciation of our work.

Reviewer 2 Report

Comments and Suggestions for Authors

The review by Tonon et al. albeit not so original (several reviews already dealt with circulating markers in liver cancers, especially circRNA and miRNA), is comprehensive, up-to-dated and well focused on the topic. 

- English language needs to be revised by a native speaker, since several typos and incorrect grammar constructions are present, especially in the introduction.

- Page 3, raws 120-121. "The combined use of AFP-L3, AFP and DCP with patient gender and age resulted in an improvement in sensitivity for HCC detection but an increase in false-positive results." The meaning of this sentence is somewhat nebulous: please remove it or develop the concept

- In miRNA paragraph as well as table 1, please consider that some miRNAs are under study for their role in the response to Sorafenib therapy [see doi: 10.1186/s13046-023-02718-w; doi: 10.1158/0008-5472.CAN-19-0472.; and others].

Comments on the Quality of English Language

English language needs to be revised by a native speaker. There are several typos and incorrect grammatical constructions, especially in the Introduction and the first paragraphs.

Author Response

The review by Tonon et al. albeit not so original (several reviews already dealt with circulating markers in liver cancers, especially circRNA and miRNA), is comprehensive, up-to-dated and well focused on the topic.

We wish to thank the reviewer for the appreciation of our work.

- English language needs to be revised by a native speaker, since several typos and incorrect grammar constructions are present, especially in the introduction.

The revised manuscript has been revised by the professional language editing, MDPI Author Services. (certification attached to the revision)

- Page 3, raws 120-121. "The combined use of AFP-L3, AFP and DCP with patient gender and age resulted in an improvement in sensitivity for HCC detection but an increase in false-positive results." The meaning of this sentence is somewhat nebulous: please remove it or develop the concept.

In the revised manuscript, the sentence has been removed.

- In miRNA paragraph as well as table 1, please consider that some miRNAs are under study for their role in the response to Sorafenib therapy [see doi: 10.1186/s13046-023-02718-w; doi: 10.1158/0008-5472.CAN-19-0472.; and others].

In the revised manuscript, the two references have been added on page 5 line 205 and the references numbering in the text, in the tables and in the final bibliography have been updated.

English language needs to be revised by a native speaker. There are several typos and incorrect grammatical constructions, especially in the Introduction and the first paragraphs.

See above

Round 2

Reviewer 2 Report

Comments and Suggestions for Authors

The Authors fully addressed all issues.